# Transcriptional subtypes on immune microenvironment and predicting postoperative recurrence and metastasis in human pheochromocytoma and paraganglioma

**Yang Liu[1], Xu Yan[2], Yibo Zhang[3], Zhenfu Gao[1], Fengrui Nan[1], Siyu Shi[1], Jingyun Chen[1], Lingyu Li[1,4]***

[1]Cancer Institute, The First Hospital of Jilin University, Changchun, China; [2]Pathology Department, The First Hospital of Jilin University, Changchun, China; [3]Clinical Laboratory, The First Hospital of Jilin University, Changchun, China; [4]Cancer Center, The First Hospital of Jilin University, Changchun, China

## eLife Assessment

This is a **valuable** study describing transcriptome-based pheochromocytoma and paraganglioma (PPGL) subtypes and exploring the mutations, immune correlates and disease progression of cases in each subtype. The cohort is a reasonable size and a second cohort is included from the Cancer Genome Atlas (TCGA). One of the key premises of the study is that identification of driver mutations in PPGL is not complete and that compromises characterisation for prognostic purposes. This is a **solid** starting point on which to base characterisation using different methods.

*For correspondence:
lilingyu@jlu.edu.cn

**Abstract** Human pheochromocytomas and paragangliomas (PPGLs) exhibit substantial molecular and immune heterogeneity, complicating risk assessment and treatment. Here, we define three distinct tumor transcriptional subtypes (C1, C2, and C3) in a clinically annotated cohort of PPGL patients through integrative transcriptomic and immunogenomic profiling. C1 is characterized by hypoxia-driven pathways and an immunosuppressive microenvironment, correlating with poor prognosis. C2 exhibits a highly inflamed immune landscape with robust CD8+ T cell infiltration, suggesting potential sensitivity to immunotherapy. C3 is enriched in metabolic reprogramming pathways and displays intermediate clinical outcomes. Genetic analysis reveals subtype-specific mutational patterns, with pseudohypoxic driver mutations (*SDHB*, *VHL*, *SDHA*, and *SDHD*) predominant in C1 and C3, while kinase pathway alterations (*NF1* and *RET*) define C2. Single-nucleus RNA sequencing of human PPGL tumors further delineates immune ecosystem diversity. Notably, we identify *ANGPT2*, *PCSK1N*, and *GPX3* as key subtype-specific biomarkers, with *ANGPT2* driving tumor progression in C1 and emerging as a potential therapeutic target. Our findings provide a refined molecular classification integrating immune and genomic features in human PPGLs, offering a framework for improved prognostication and precision therapies in this rare neuroendocrine tumor type.

## Introduction

Pheochromocytomas and paragangliomas (PPGLs) are rare, highly heterogeneous neuroendocrine (NE) tumors that pose significant challenges in prognosis assessment due to their complex molecular landscape and variable clinical behavior (*Dahia, 2017*; *Calsina et al., 2023*). Postoperative recurrence and metastasis rates for PPGLs vary widely, ranging from 6.0 to 17.4% (*Parisien-La Salle et al., 2022*), complicating treatment and management strategies. An accurate assessment of the risk for recurrence and metastasis following surgery is essential for optimal management, as it directly informs decisions regarding postoperative surveillance and therapy.

Current genomic subtypes, such as pseudohypoxic, kinase signaling, and Wnt-altered subtypes, have shown promise in identifying the molecular characteristics of PPGLs (*Burnichon et al., 2011*). However, these approaches have limited applicability, as the overall germline mutation rate in PPGL patients is only 30–40% (*Horton et al., 2022*; *Amar et al., 2005*), leaving a substantial portion of PPGLs without actionable genomic markers. This creates a gap in our ability to predict recurrence and metastasis based solely on genetic profiles. While some genotype–phenotype correlations have been established (*Cui et al., 2021*; *Zethoven et al., 2022*; *Li et al., 2024*), molecular subclassifications have advanced our understanding of PPGL biology, the role of secondary molecular events, particularly in the tumor microenvironment (TME), remains underexplored. The TME has been shown to influence prognosis and response to treatment in other tumor types, and emerging evidence suggests that it may play a critical role in the development and progression of PPGLs. However, its precise contribution to disease progression in PPGLs has yet to be fully elucidated.

This study aims to bridge these gaps by analyzing multiple PPGL cohorts with a focus on the transcriptional landscape and immune microenvironment. Using weighted gene co-expression network analysis (WGCNA) (*Langfelder and Horvath, 2008*), we seek to define the transcriptional subtypes of PPGLs, explore how these subtypes are linked to immune infiltration patterns, and identify key transcriptional alterations associated with recurrence and metastasis. We also aim to propose potential prognostic markers and assess the impact of immune microenvironment profiles on disease progression. This comprehensive analysis will provide new insights into the biology of PPGLs and improve the predictive accuracy of postoperative recurrence and metastasis risk.

## Results

### Characterization of three transcriptional subtypes in PPGLs via WGCNA

In this study, we employed WGCNA to explore the transcriptional landscape of PPGLs, successfully identifying three distinct transcriptional subtypes that are associated with varying clinical outcomes and may affect disease progression through mechanisms such as immune infiltration. Our analysis encompassed a comprehensive cohort of 87 PPGL patients from the First Hospital of Jilin University (PPGL cohort), supplemented by an extensive dataset from The Cancer Genome Atlas (TCGA) that included 179 samples (TCGA cohort), which allowed us to validate our findings on a broader scale. We utilized immunohistochemistry (IHC), RNA sequencing, and whole-genome sequencing to analyze the samples, integrating molecular profiles with clinical data to predict recurrence and metastasis based on the identified transcriptional subtypes (*Figure 1A*, *Figure 1—source data 1*). We harmonized anatomic site annotations from our PPGL cohort and the TCGA cohort. Stacked bars depict the proportion of adrenal pheochromocytomas (PC, red) and extra-adrenal paragangliomas (PG, blue) within each subtype (*Figure 1—figure supplement 1*). The site distribution is essentially the same across subtypes—approximately three-quarters PC and one-quarter PG—with only minimal, non-systematic variation. Using WGCNA on the PPGL cohort, we resolved 20 co-expression gene modules. Module-level functional enrichment (*Figure 1B*) grouped these modules into two overarching biological programs—metabolism-associated pathways and hypoxia-inducible factor (HIF)-1 signaling/proliferation—both central to tumor biology and immune modulation. Clustering tumors by these program signatures delineated three major transcriptional subtypes, designated Cluster 1 (C1), Cluster 2 (C2), and Cluster 3 (C3). Subtype C1 is enriched in pathways associated with hypoxia and HIF-1 signaling, which are vital for tumor survival and aggressiveness (*Domènech et al., 2021*). Subtype C3's enrichment in metabolic pathways, such as fatty acid metabolism and parathyroid metabolism, reflects its adaptation to metabolic stress, potentially revealing targets for metabolic intervention.

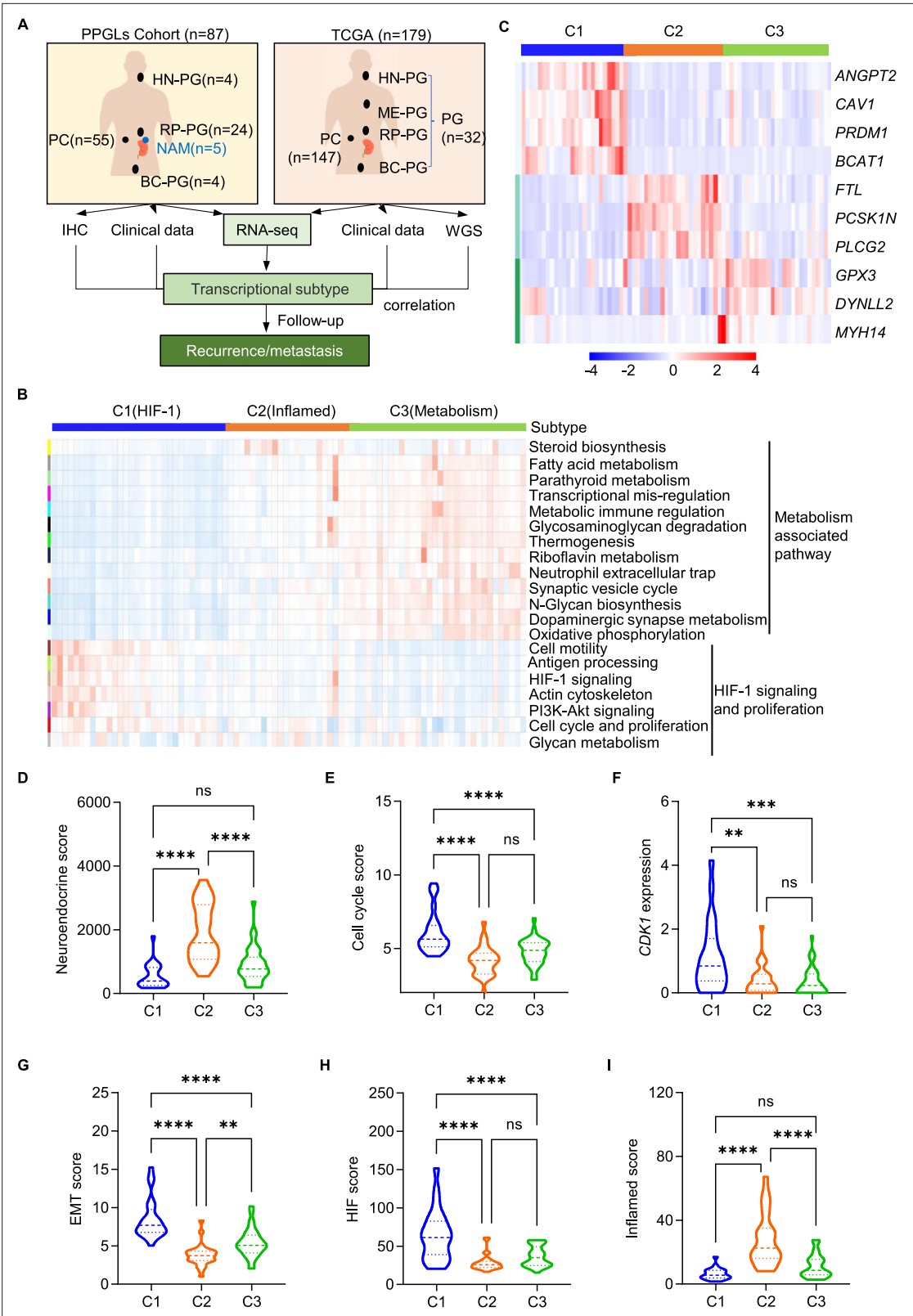

**Figure 1.** Divergence in transcriptional subtypes defined three PPGL subtypes across different datasets. (**A**) Overview of PPGL cohort (*n* = 87) and TCGA cohort (*n* = 179) detailing the number of cases by anatomical localization: HN-PG (head and neck paraganglioma), ME-PG (mediastinal paraganglioma), RP-PG (retroperitoneal paraganglioma), BC-PG (bladder paraganglioma), and PC (pheochromocytoma). Clinical data, whole-genome sequencing (WGS), and RNA-seq were used for subtype identification and correlation analysis related to recurrence/metastasis. (**B**) Heatmap

*Figure 1 continued on next page*

*Figure 1 continued*

depicting key metabolic and biological pathways associated with each subtype and displaying the pathway enrichment analysis for three subtypes of pheochromocytoma: C1 (HIF-1), C2 (inflamed), and C3 (metabolism). Each row represents a specific biological pathway, and the color intensity represents the degree of enrichment across the subtypes. (**C**) Heatmap showing gene expression changes across subtypes. Specific genes like *ANGPT2* and *CAV1* are upregulated in C1 compared to other subtypes. (**D, E**) Violin plots displaying the enrichment scores for Neuroendocrine and cell cycle among the three subtypes (C1, C2, and C3). (**F**) Violin plots displaying *CDK1* among the three subtypes (C1, C2, and C3). (**G–I**) Violin plots displaying the enrichment scores for epithelial–mesenchymal transition (EMT), HIF, and inflamed among the three subtypes (C1, C2, and C3). Comparisons were calculated by one-way ANOVA (**D–I**). Data are presented as mean ± SD. ns, $p > 0.05$; **$p < 0.01$; ***$p < 0.001$; ****$p < 0.0001$.

The online version of this article includes the following source data and figure supplement(s) for figure 1:

**Source data 1.** PPGLs samples information.

**Figure supplement 1.** Distribution of tumor site categories—pheochromocytoma (PC) vs paraganglioma (PG)—by transcriptional subtype (C1–C3) in the aggregated dataset (TCGA + Magnus).

**Figure supplement 2.** Validation of PPGL transcriptional subtype scores in TCGA and Magnus cohorts.

A detailed analysis of gene expression within each subtype revealed several critical marker genes that characterize each group (*Figure 1C*). Subtype C1 is marked by genes involved in HIF-1 signaling and angiogenesis, such as *ANGPT2* and *CAV1* (*Kapiainen et al., 2021*; *Nwosu et al., 2016*). In contrast, subtype C2 is defined by genes, including *PCSK1N* and *FTL*, while subtype C3 shows a significant upregulation of genes like *DYNLL2* and *GPX3*. The NE features, assessed through an NE score, reveal distinct patterns among the three PPGL subtypes (*Figure 1D*). Notably, subtype C2 demonstrates significantly elevated NE score in comparison to subtypes C1 and C3, suggesting a strong NE phenotype. Subtype C1 exhibited higher cell cycle scores (*Figure 1E*), expression of CDK1 (*Figure 1F*; *Calsina et al., 2023*), epithelial–mesenchymal transition (EMT) scores (*Figure 1G*), and HIF signaling scores (*Figure 1H*) compared to subtypes C2 and C3. This suggests that C1 may promote rapid tumor growth, metastasis, and potentially lead to a poorer prognosis. Subtype C2, on the other hand, shows an increased inflamed score compared to subtypes C1 and C3, suggesting a more inflamed TME that may be more responsive to immunotherapy (*Figure 1I*).

We aggregated the snRNA-seq data to the sample level by summing gene counts across nuclei to generate pseudo-bulk profiles. These profiles were then normalized for library size, log-transformed (log1p), and *z*-scaled across samples. Using gene-set scores derived from our WGCNA analysis of PPGLs, we subsequently defined subtypes in the TCGA cohort (*Figure 1—figure supplement 2A*) and in Magnus's cohort (*Figure 1—figure supplement 2B*), revealing distinct transcriptional subtypes with unique molecular features that underscore disease heterogeneity and suggest mechanisms that may drive divergent clinical behaviors.

## Genetic mutation patterns and pathway alterations across transcriptional subtypes of PPGLs

To elucidate the relationship between transcriptional subtypes and previous genetic mutation classifications, we utilized 84 variation dates from 179 PPGL patients in TCGA cohort and 30 PPGL patients in Magnus's cohort. The mutation profile for each subtype highlights distinct genetic alterations in PPGLs (*Figure 2A*, *Figure 2—source data 1*). Subtype C1 exhibits a high frequency of mutations in genes associated with the pseudohypoxia pathway, such as *VHL* and *SDHB* (*Crona et al., 2019*), suggesting a unique adaptation to hypoxic conditions that may drive tumor progression. In contrast, subtype C2 predominantly features mutations in genes involved in kinase signaling pathways, including *NF1* and *RET* (*Fishbein et al., 2017*), which imply active signaling networks that could influence tumor growth and response to therapies. Subtype C3 does not exhibit a clearly defined pathway but presents a diverse mutational landscape, indicating the complexity of tumor biology in this group.

When analyzing the proportion of genetic alterations in key pathways (*Figure 2B*), we found that more than half of the cases in each subtype lacked detectable driver mutations. Among the patients with genetic alterations, subtypes C1 and C3 exhibited a significantly higher proportion of pseudohypoxic signaling alterations compared to C2. In contrast, subtype C2 was more strongly enriched for kinase pathway mutations and WNT alterations. A closer examination of pseudohypoxic mutations revealed a significant concentration of *SDHB* and *VHL* mutations in subtype C1, while subtype C3 exhibited a notable concentration of *SDHA* and *SDHD* mutations (*Figure 2C*). Notably, subtype C2

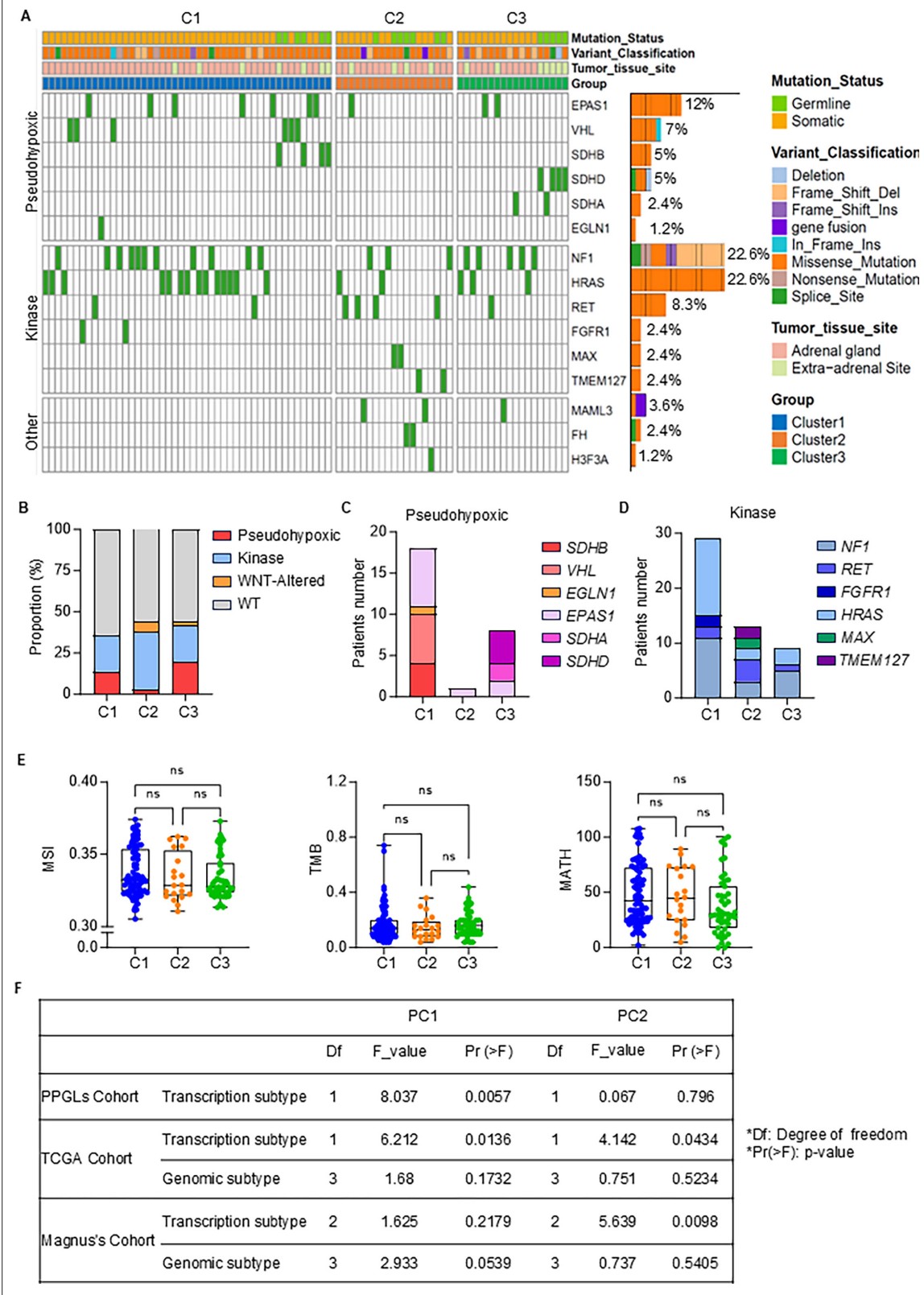

**Figure 2.** Mutational landscape and molecular characterization of PPGL cohorts. (**A**) Overview of the genomic alterations across the three subtypes (C1, C2, and C3) in TCGA cohorts. Genes associated with pseudohypoxic, kinase, and other pathways are shown, with individual rows representing distinct gene mutations and columns corresponding to samples. Bar plot showing the proportional distribution of variant classifications. (**B**) Proportional analysis of the transcriptional subtypes across C1, C2, and C3. The pseudohypoxic, kinase, WNT-altered, and wild type (WT) subtypes (lacking pathogenic

*Figure 2 continued on next page*

*Figure 2 continued*

mutations) are depicted, illustrating the distribution of each subtype. (**C**) The number of patients with mutations in the pseudohypoxic-related genes (*SDHB*, *VHL*, *EGLN1*, *EPAS1*, *SDHA*, and *SDHD*) across the three subtypes. (**D**) The number of patients with mutations in kinase-related genes (*NF1*, *RET*, *FGFR1*, *HRAS*, *MAX*, and *TMEM127*) across subtypes. (**E**) Comparisons of microsatellite instability (MSI), tumor mutation burden (TMB), and MATH (mutant allele tumor heterogeneity) scores across the three subtypes. (**F**) Linear regression analysis of beta coefficients (PC1 and PC2) for different cohorts including PPGL cohort, TCGA, and Magnus's cohort. Comparisons were calculated by one-way ANOVA (**E**). Data are presented as mean ± SD. ns, $p > 0.05$.

The online version of this article includes the following source data for figure 2:

**Source data 1.** TCGA+SNRNA_mutation.

patients exhibited the highest proportion of kinase pathway involvement, along with the greatest variety of kinase pathway mutations (*Figure 2D*).

Mutation features, such as microsatellite instability (MSI), tumor mutational burden (TMB), and mutations in pathways like MYC-associated factor X (MAX), which are commonly used to assess the efficacy of clinical immunotherapy, do not show significant differences among subtypes (*Figure 2E*). Additionally, principal component analysis (PCA) effectively distinguished transcriptional subtypes within the PPGL cohort, further supporting the validity of the subtype classification (*Chu et al., 2024*). In both the Magnus (*Zethoven et al., 2022*) and TCGA cohorts, PCA similarly confirmed that transcriptional subtypes could significantly differentiate PPGL subgroups, and that transcriptional subtyping outperformed traditional genetic mutation-based classification in distinguishing PPGLs (*Figure 2F*). In summary, our study offers a thorough molecular characterization of PPGLs, identifying three distinct subtypes with unique transcriptional features.

## Immune infiltration profiles across transcriptional subtypes of PPGLs

To elucidate the immune infiltration characteristics of different transcriptomic subtypes, we applied the xCell (*Aran et al., 2017*) algorithm to analyze the RNA-seq data from the PPGLs and TCGA cohorts, revealing TME infiltration features at the transcriptional level. The heatmap illustrates the distribution and abundance of different immune cells across the three subtypes (*Figure 3A*). Subtype C2, which showed significantly higher inflamed scores compared to subtypes C1 and C3 (*Figure 1I*), exhibited increased levels of CD4$^+$ Th1 cells and cytotoxic lymphocytes, such as T cells and natural killer (NK) cells. In contrast, hematopoietic stem cells (HSCs) and endothelial cells were found to be reduced. IHC staining analysis of infiltrated CD8$^+$ T cells further supports the quantitative findings from the inflamed score, demonstrating that subtype C2 has more pronounced CD8$^+$ T cell infiltration at the tumor site compared to subtypes C1 and C3 (*Figure 3B*). These findings reinforce the heatmap data and suggest an active cytotoxic immune response in C2. In further statistical analysis, both the PPGL cohort and the TCGA cohort showed that the C2 subtype had significantly higher infiltration of cytotoxic lymphocytes and CD4$^+$ Th1 cells, while exhibiting low infiltration of immunosuppressive cells (such as HSC), reflecting an activated immune microenvironment. In contrast, the C1 subtype was enriched in HSC but showed the lowest levels of cytotoxic lymphocyte and CD4$^+$ Th1 cell infiltration. The immune microenvironment infiltration profile of the C3 subtype was intermediate between the two (*Figure 3C, D*). This validation in the TCGA cohort, a larger, independent dataset, underscores the robustness of our findings and their potential relevance across different PPGL populations.

To further investigate the relationship between transcriptomic subtypes and the presence of tumor-infiltrating cytotoxic lymphocytes (TILs), we applied multiple linear regression models. After adjusting for various potential factors that could influence TIL infiltration, we aimed to determine whether transcriptomic subtypes still have an impact on the degree of TIL infiltration. In the PPGL cohort, regardless of whether the crude model (Model 1), the age, gender, and primary site-adjusted Model 2, or Model 3 (which further adjusted for Ki-67, SSTR2, and SDHB expression based on Model 2) was used, the results showed that compared to the subtype C1, the subtype C2 exhibited the most significant TILs infiltration, followed by the subtype C3 (*Figure 3E*). This conclusion was also validated in the TCGA cohort (*Figure 3F*). In summary, our extensive immunological profiling across the transcriptional subtypes of PPGLs reveals distinct immune landscapes, with subtype C2 exhibiting an active immunological profile.

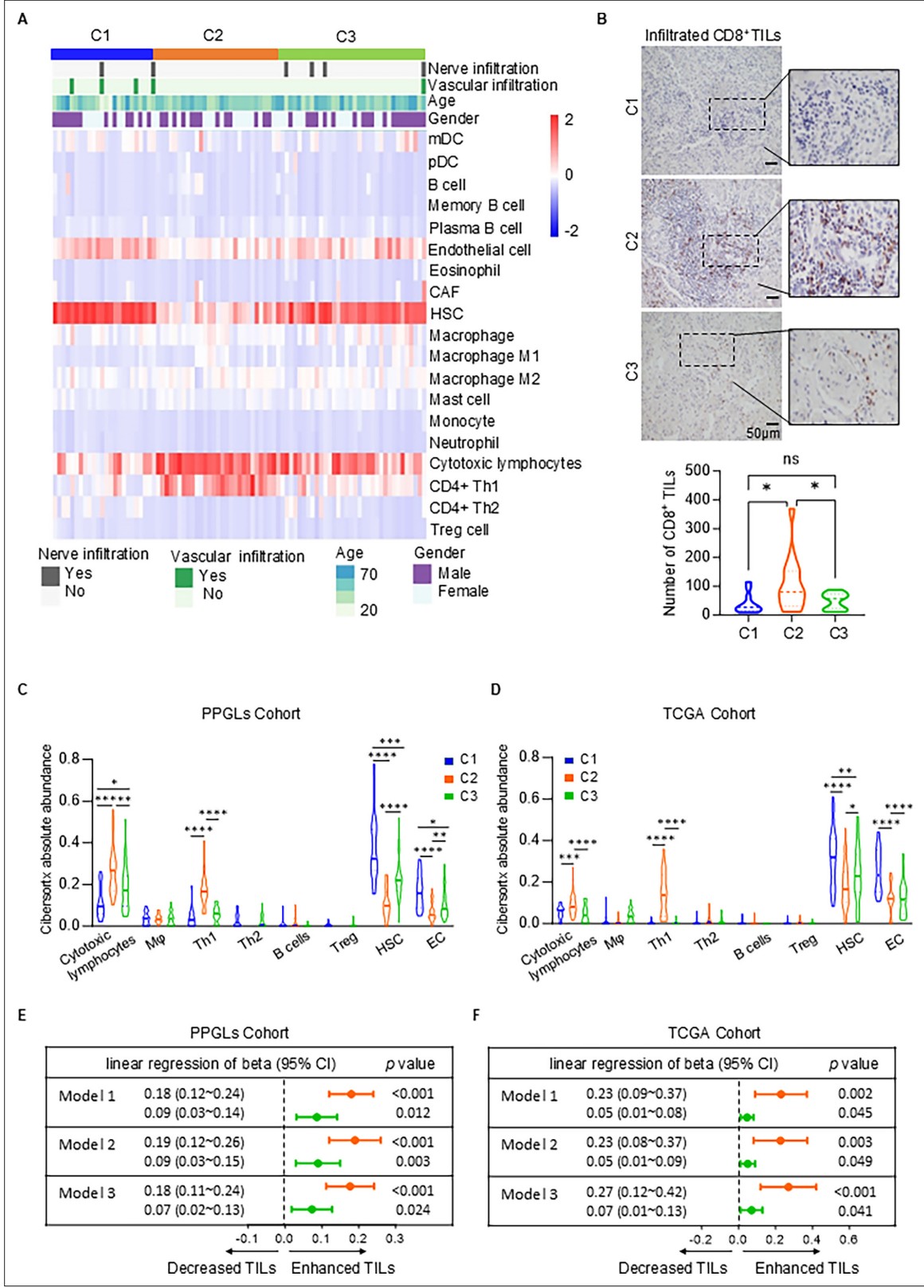

**Figure 3.** Immune infiltration landscape across PPGL and TCGA cohorts. (**A**) Heatmap displaying the immune cell composition and clinical features, including nerve and vascular infiltration, gender, and age across C1, C2, and C3 subtype. The heatmap highlights variations of immune cell infiltration abundance across different samples in immune cell types such as dendritic cells (DC), macrophages, B cells, and natural killer (NK) cells. (**B**) Representative immunohistochemistry images (top) and the absolute number (bottom) of infiltrated CD8[+] T cells in C1, C2, and C3 subtypes. The

*Figure 3 continued on next page*

*Figure 3 continued*

infiltrated number of CD8$^+$ tumor-infiltrating lymphocytes (TILs) was determined by randomly selecting 10 fields at 40×magnification and calculating the total count of CD8$^+$ cells. The xCell analysis in the PPGL cohort (**C**) and TCGA cohort (**D**) depicting absolute abundance of immune cells across C1, C2, and C3 subtype. Th1, T helper 1 cells; Th2, T helper 2 cells; Mφ, macrophages; HSC, hematopoietic stem cell; EC, endothelial cell. Cytotoxic lymphocytes include both NK and T cells. The forest plot illustrates tumor-infiltrating cytotoxic lymphocytes (TILs) across different transcriptional subtypes in the PPGL cohort (**E**) and the TCGA cohort (**F**), using linear regression models. Model 1 was a crude model; Model 2 was adjusted for age, gender, and primary tumor location based on Model 1; Model 3 was adjusted for *MKI67*, *SDHB*, and *SSTR2* expression based on Model 2. Each model is compared to subtype C1 as the reference group. Orange represents subtype C2, and green represents subtype C3. Comparisons were calculated by one-way ANOVA (**B–D**). Data are presented as mean ± SD. Ns, $p > 0.05$; *$p < 0.05$; **$p < 0.01$; ***$p < 0.001$; ****$p < 0.0001$.

## Transcriptional diversity and immune microenvironment characteristics in PPGL subtypes

To further clarify the TME characteristics of each transcriptional subtype, we performed single-nucleus RNA sequencing (snRNA-seq) analysis on 32 PPGL patients in Magnus's cohort (*Zethoven et al., 2022*). The UMAP analysis successfully categorized the genetic profiles into three distinct subtypes: C1, C2, and C3, while also identifying various cell types, including immune cells, chromaffin, myeloid, and fibroblast cells across the samples (*Figure 4A*). This diverse cellular representation highlights the complex biological composition of PPGLs and correlates with specific mutational profiles such as *SDHB*, *VHL*, and *RET*. Consistent with transcriptomic sequencing data from the TCGA cohort (*Figure 2A–D*), subtype C1 predominantly harbors *VHL* and *SDHB* mutations, while subtype C2 is primarily enriched in mutations related to kinase pathways such as *RET* and *HRAS*. Subtype C3 is mainly characterized by mutations in *SDHA* and *SDHD*. After extracting the immune cells, we performed further cluster analysis. UMAP visualization revealed that the infiltrating immune cells were categorized into 12 distinct subsets, including CD8$^+$ T, CD4$^+$ T, NK, B cells, dendritic cells (DC), and others (*Figure 4B*). Further examination of immune cell distribution using UMAP in subtypes C1, C2, and C3, compared to normal tissue, reveals distinct clustering patterns of immune cells (*Figure 4C, D*). This visualization indicates a higher proportion of cytotoxic lymphocyte immune infiltration, specifically CD8$^+$ T cells, in subtype C2 compared to subtypes C1 and C3, with no significant involvement of NK cells. Although no significant differences were observed in the overall macrophage content analysis across the different transcriptional subtypes (*Figure 2C, D*), further clustering of macrophages using single-cell sequencing revealed an increased proportion of tumor-suppressing M1 macrophages and a significantly reduced proportion of tumor-promoting tumor-associated macrophages (TAMs) in the C2 subtype compared to the other two subtypes.

To determine whether the increased CD8$^+$ T cell infiltration in subtype C2 is associated with enhanced immunoregulatory function, we analyzed ligand–receptor interactions using CellChat. The results revealed that the interactions between CD8$^+$ T cells and themselves, as well as with CD4$^+$ T cells and NK cells, were significantly enhanced through chemokine pathways such as CCL–CCR. In contrast, the TGFB–TGFBR ligand–receptor interaction between CD8$^+$ T cells and immunosuppressive M2 macrophages and TAM cells was significantly weakened (*Figure 4E*). In conclusion, the increased infiltration of CD8$^+$ T cells in subtype C2 may contribute to immune activation by interacting with other immune cells in the microenvironment.

Pseudotime analysis was used to elucidate the developmental pathways of tumor cells in PPGLs, highlighting dynamic shifts between transcriptional states, particularly the transition from a naive to an activated state within the TME (*Qiu et al., 2017*). Pseudotime analysis reveals that compared with normal adrenal cells, subtypes C2 and C1 follow distinct developmental trajectories, while subtype C3 occupies an intermediate position between the C1 and C2 pathways. This suggests that C3 may represent a transitional state between the two, which helps explain the significant differences observed between C1 and C2 in terms of NE characteristics, tumor proliferation, HIF-1 pathway activity, and the tumor immune microenvironment, with C3 consistently lying between them (*Figure 4F*).

In summary, our multi-omics analysis illuminates the transcriptional diversity and intricate immune interactions present in PPGLs, providing a comprehensive overview of the genetic and immune profiles that characterize each subtype. These results not only deepen our understanding of PPGL biology but also open avenues for more targeted and effective therapeutic strategies tailored to the distinct features of each subtype.

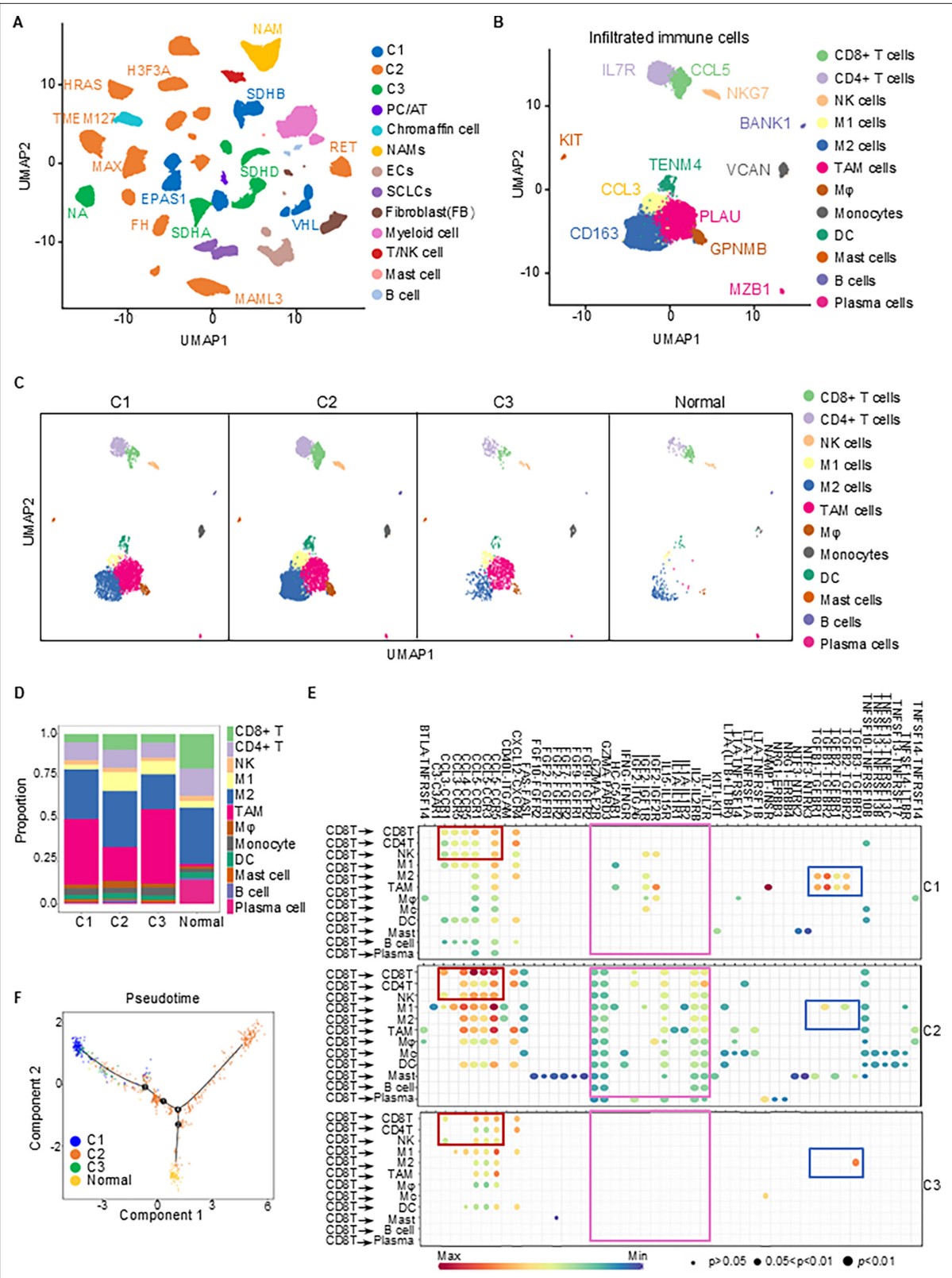

**Figure 4.** UMAP analysis of cell types and pseudotime trajectories in different subtypes. (**A**) UMAP plot showing the distribution of various cell types across three subtypes (C1, C2, and C3) and normal tissues. Each subtype is uniquely colored, with consistent coloring for the same subtype. Corresponding mutated genes are indicated. PC/AT, pheochromocytoma/adrenal tumor; NAMs, neuroblastic cells; ECs, endothelial cells; SCLCs, Schwann cell-like cells. (**B**) UMAP plot highlighting specific immune cells across three subtypes and normal tissues, with corresponding marker genes

*Figure 4 continued on next page*

*Figure 4 continued*

indicated. (**C**) UMAP plot highlighting specific immune cells in each subtype: C1, C2, C3, and normal tissues. (**D**) Bar plot represents the proportion of each immune cell type in the C1, C2, C3, and normal subtypes. (**E**) The dot plot represents a ligand–receptor interaction plot generated using CellChat, showcasing the communication network between various immune cell types across three different subtypes (C1, C2, and C3). The rows represent different immune cell populations, while the columns depict various ligands and receptors involved in cell–cell communication. The color gradient from red to green indicates interaction strength, with red representing stronger interactions and green representing weaker ones. The size of the dots reflects the significance level of each interaction. Distinct regions of enhanced interactions are highlighted with colored boxes, emphasizing subtype-specific signaling patterns. (**F**) Pseudotime analysis revealing the developmental trajectories of cell populations in different subtypes. Arrows indicate the progression paths of cell differentiation, suggesting different developmental stages and lineage decisions across C1, C2, C3, and normal subtype.

## Subtype-specific gene expression profiles and clinical outcomes in PPGLs

To convert this transcriptional subtype into a clinically applicable classification tool, we initially employed a linear regression model to compare the effect values ($\beta$ value) of various candidate marker genes (*Figure 5—figure supplement 1A–C*) on the transcriptional subtypes. We then identified the genes with the most significant effect values and statistical differences for each subtype as the marker genes for that subtype. Ultimately, we identified the genes *ANGPT2*, *PCSK1N*, and *GPX3*, which are significantly overexpressed and have the most significant β values in subtypes C1, C2, and C3, respectively, as the marker genes for these three subtypes (*Figure 5A*, *Figure 5—figure supplement 1A–C*). *ANGPT2*, involvement in angiogenesis and vascular remodeling (*Kapiainen et al., 2021*), had significantly higher expression in subtype C1 compared to C2 and C3, which may contribute to the aggressive characteristics of this subtype. *PCSK1N*, encoding the NE protein ProSAAS (*Feng et al., 2002*), exhibited notably high expression levels in subtype C2. This is consistent with our finding of significantly elevated NE scores in subtype C2 (*Figure 1D*), further suggesting that this subtype possesses the most prominent NE characteristics. Additionally, *GPX3*, a significantly upregulated molecule in the C3 subtype, is involved in the regulation of redox homeostasis (*Kritsiligkou et al., 2017*), which may be closely associated with the active metabolic state in this subtype. IHC staining confirmed the differential expression patterns of ANGPT2, PCSK1N, and GPX3 in C1, C2, and C3 subtypes, respectively (*Figure 5B*). These findings not only validate the transcriptomic data but also emphasize the potential biological roles these genes may play in the pathophysiology of PPGLs.

A Kaplan–Meier survival curve analysis revealed significant differences in patient outcomes based on subtype classification (*Figure 5C*). Patients in subtype C1 exhibited markedly worse progression-free survival (PFS) compared to those in subtypes C2 and C3, underscoring the critical role of subtype stratification in predicting clinical outcomes. Additionally, Cox regression analysis quantified the impact of these subtypes on PPGL patient prognosis (*Figure 5D*). The models, adjusted for various factors, confirmed that subtype C1 was associated with the highest risk of recurrence and metastasis. In contrast, subtype C2 demonstrated the most favorable PFS, significantly outperforming both subtype C1 and subtype C3.

In conclusion, our analysis highlights the potential of *ANGPT2*, *PCSK1N*, and *GPX3* as subtype-specific biomarkers, offering valuable insights into the molecular heterogeneity and clinical prognosis of PPGLs.

## *ANGPT2* as a key regulator and diagnostic marker in PPGL subtype C1

*ANGPT2* had significantly higher expression in subtype C1 compared to C2 and C3, indicating its potential involvement in angiogenesis and vascular remodeling, which may contribute to the aggressive characteristics of this subtype. The receiver operating characteristic (ROC) curve highlights the diagnostic potential of *ANGPT2* in subtype C1, with an area under the curve of 0.880 (*Figure 6A*), demonstrating its high sensitivity and specificity in distinguishing between the subtypes based on their molecular profiles. The samples were categorized into *ANGPT2*^high and *ANGPT2*^low groups based on the cutoff of *ANGPT2*'s ROC in the PPGL cohort. It was revealed that the *ANGPT2*^high group was associated with higher scores in the cell cycle, EMT, and HIF-1 signaling pathways (*Figure 6B–D*). This suggests that *ANGPT2* may not only serve as a specific biomarker for subtype C1 but also act as a regulator of tumor progression and the tumor's ability to adapt to hypoxic conditions.

To further validate our findings, we employed the rat PPGL cell line PC12 and used CRISPR–Cas9 technology to knock out (KO) the *ANGPT2* gene, which was confirmed by western blot

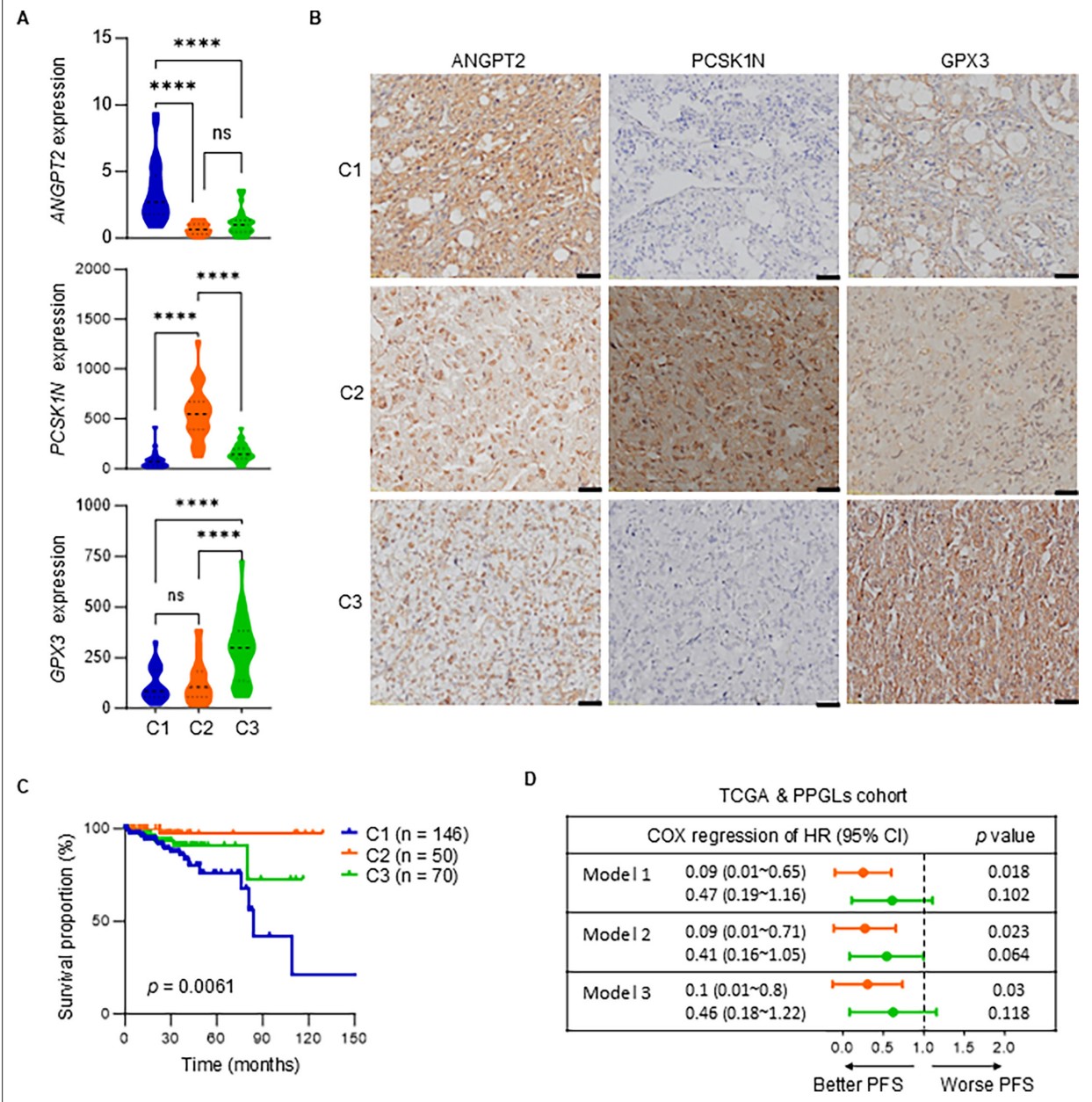

**Figure 5.** The specific genetic features of different PPGL subtypes. (**A**) Violin plots depicting the expression levels of *ANGPT2*, *PCSK1N*, and *GPX3* gene across three identified subtypes in PPGL cohort. (**B**) Immunohistochemistry images showing the protein expression of ANGPT2, PCSK1N, and GPX3 across subtypes C1, C2, and C3. Scale bars represent 50 μm. (**C**) Survival curves comparing the progression-free survival (PFS) of patients across three subtypes of PPGLs, integrating data from both the PPGL and TCGA cohorts. (**D**) The forest plot displaying the hazard ratios (HR) with 95% confidence intervals (CI) for PFS across PPGL subtypes using Cox regression analysis. Model 1 was a crude model; Model 2 was adjusted for age, gender, and race based on Model 1; Model 3 was further adjusted for tumor location and pathogenic mutation type based on Model 2. Each model is compared to subtype C1 as the reference group. Orange represents subtype C2, and green represents subtype C3. Comparisons were calculated by one-way ANOVA (**A**). Data are presented as mean ± SD. ns, $p > 0.05$; ****$p < 0.0001$.

The online version of this article includes the following figure supplement(s) for figure 5:

**Figure supplement 1.** Effect sizes of marker genes for PPGL transcriptional subtype classification.

analysis (*Figure 6E*). We then compared the tumor growth curves of *ANGPT2* KO (*ANGPT2$^{-/-}$*) and *ANGPT2$^{WT}$* PC12 cells in subcutaneous xenograft models in BALB/c nude mice. The results showed that the in vivo growth of *ANGPT2$^{-/-}$* cells was significantly slower (*Figure 6F, G*). RNA-seq analysis confirmed that, after *ANGPT2* knockout, the expression scores of key genes involved in EMT, tumor

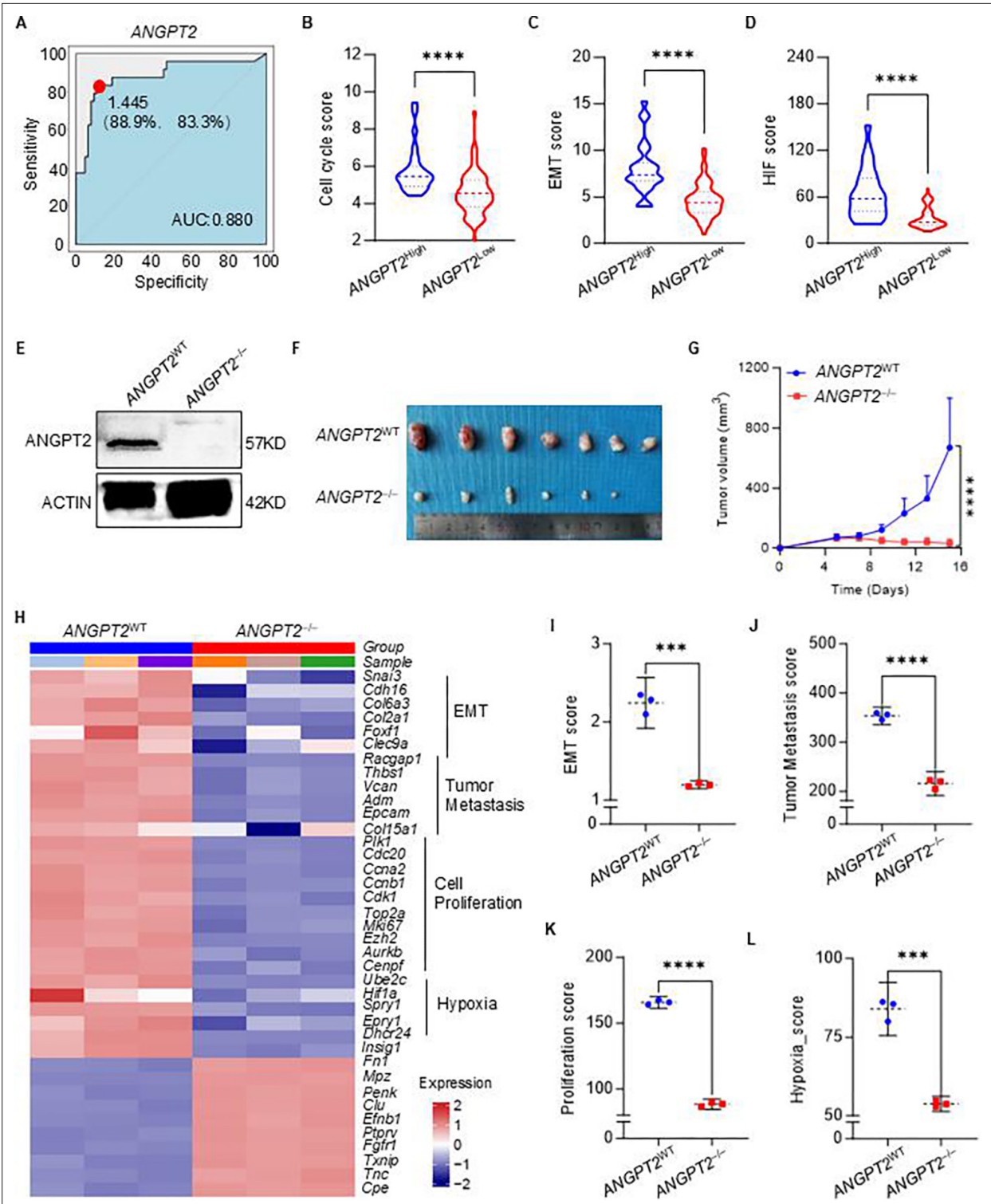

**Figure 6.** The characteristics of *ANGPT2* knockout. (**A**) The ROC curve illustrates the diagnostic ability to distinguish *ANGPT2* expression in PPGLs, specifically differentiating subtype C1 from non-C1 subtypes. The red dot indicates the point with the highest sensitivity (88.9%) and specificity (83.3%). AUC, the area under the curve. (**B–D**) Violin plots display the distribution of cell cycle scores, epithelial–mesenchymal transition (EMT) scores, and HIF scores in PPGLs between *ANGPT2*high and *ANGPT2*low Groups, using a cutoff value of 1.445 for *ANGPT2* expression as shown in panel A. (**E**) Immunoblots showing ANGPT2 and Actin expression in *ANGPT2* wild-type (*ANGPT2*WT) and knockout (*ANGPT2*−/−) rat PPGL cell line PC12. (**F**) The graph shows a series of tumors harvested from BALB/c nude mice xenografted with either *ANGPT2*WT or *ANGPT2*−/− PC12 tumors over 16 days. (**G**) The tumor growth volume of *ANGPT2*WT or *ANGPT2*−/− PC12 tumors in Balb/c nude mice. Data are presented as mean ± SD (n=7 mice per group). Statistical significance

*Figure 6 continued on next page*

*Figure 6 continued*

was assessed by unpaired two-tailed Student's t-test. (**H**) Heatmap illustrates changes in gene expression related to cell proliferation, tumor metastasis, and hypoxia between *ANGPT2*^WT or *ANGPT2*^−/− PC12 cells. (**I–L**) The dotplots illustrate EMT scores, tumor metastasis scores, cell proliferation scores, and hypoxia scores in *ANGPT2* KO tumors. Comparisons were calculated by t tests (**B–D, I–L**). Data are presented as mean ± SD. \*\*\*$p < 0.001$; \*\*\*\*$p < 0.0001$.

The online version of this article includes the following source data for figure 6:

**Source data 1.** Original files for western blot analysis displayed in *Figure 6E*.

**Source data 2.** Zip file containing original western blots for *Figure 6E*, indicating the relevant bands.

metastasis, proliferation, and hypoxia-related pathways were markedly reduced in the PC12 cell line (*Figure 6H–L*). This suggests that *ANGPT2*, a marker gene for subtype C1, functions as a regulatory molecule in the malignant biological behavior of PPGLs. Targeting this molecule may provide a new therapeutic approach for subtype C1 of PPGLs, which is associated with the highest malignancy and poorest prognosis.

## Discussion

The significant biological heterogeneity and wide distribution of PPGLs contribute to substantial differences in patient outcomes, making their classification and risk assessment a critical focus in clinical practice. Traditional classification methods based on anatomical origin and pan-genomic approaches centered on pathogenic mutations have offered insights but fail to fully capture the complexity of these tumors, given the relatively low prevalence of detectable mutations.

Our study addresses these gaps by introducing a novel transcriptional subtype classification framework that effectively distinguishes PPGL patients based on transcriptomic expression. This classification demonstrates significantly improved differentiation capability compared to pan-genomic classification and has been validated across independent cohorts. IHC analysis further confirmed the robustness of this classification, revealing subtype-specific clinicopathological features that closely align with their molecular characteristics, underscoring its clinical applicability.

Analysis of genomic characteristics revealed that transcriptional subtypes C1 and C3 were enriched in pseudohypoxic genomic subtypes but exhibited distinct mutation profiles: C1 was primarily associated with *SDHB* and *VHL* mutations, while C3 was predominantly linked to *SDHA* and *SDHD* mutations. Notably, more than half of the patients across all subtypes lacked detectable pathogenic mutations, yet their transcriptome expression profiles mirrored those of patients with such mutations. This finding highlights a previously overlooked patient population in genomic studies.

Further investigation into the TME revealed substantial differences among subtypes. Subtype C1 exhibited the most immunosuppressive microenvironment, characterized by low infiltration of CD8$^+$ T cells and CD4$^+$ Th1 cells, high infiltration of HSCs, and an elevated proportion of TAMs. Immune activation pathways were weaker in C1, with enhanced inhibitory interactions involving M2 macrophages and TAMs. In contrast, the C2 subtype demonstrated the most activated and inflammatory TME, with the highest inflammatory scores and robust immune cell interactions. These findings provide a foundation for stratified immunotherapy strategies.

The assessment of postoperative recurrence and metastatic risk in PPGLs remains one of the most critical challenges in current clinical practice (*Kiriakopoulos et al., 2023*). Identifying high-risk patients with precision has become a major focus of research. While several approaches, such as pathology-based scoring systems (e.g., PASS and PAGG scores) and genomic mutation-based evaluation systems (*Kim et al., 2024*), have been explored, none have achieved satisfactory outcomes.

Compared to existing clinical and molecular predictors, risk assessment in PPGL has long relied on the following indicators: clinicopathological features (e.g., tumor size, non-adrenal origin, specific secretory phenotype, and Ki-67 index), histopathological scoring systems (such as PASS/GAPP), and certain genetic alterations (including high-risk markers like SDHB inactivation mutations, as well as susceptibility gene mutations in ATRX, TERT promoter, MAML3, VHL, NF1, among others). Although these metrics are highly actionable in clinical practice, they exhibit several limitations: first, current molecular markers only cover a subset of patients, and technical constraints hinder the detection of many potentially significant variants (e.g., non-coding mutations), thereby compromising the comprehensiveness of prognostic evaluation; second, histopathological scoring is susceptible to interobserver

variability; furthermore, the lack of standardized detection and evaluation protocols across institutions limits the comparability and generalizability of results.

Our transcriptomic classification system—comprising C1 (pseudohypoxic/angiogenic signature), C2 (kinase signaling signature), and C3 (SDHx-related signature)—provides a complementary approach to PPGL risk assessment. These subtypes reflect distinct biological backgrounds tied to specific genetic alterations and can be approximated by measuring the expression of individual genes (e.g., *ANGPT2*, *PCSK1N*, or *GPX3*). This study demonstrates that the classifier offers three major advantages: first, it accurately distinguishes subtypes with coherent biological features; second, it retains significant predictive value even after adjusting for clinical covariates; third, it can be implemented using readily available assays such as IHC. These findings suggest that integrating transcriptomic subtyping with conventional clinical markers may offer a more comprehensive and generalizable risk stratification framework. However, this strategy would require validation through multi-center prospective studies and standardization of detection protocols.

In this study, subtype C1 was also identified as the poorest prognostic group, exhibiting heightened tumor proliferation, enhanced EMT, and the highest recurrence and metastatic risk. Importantly, we identified *ANGPT2* was identified as a key marker gene and regulator driving the aggressive behavior of C1, providing both a novel tool for identifying high-risk patients and a potential therapeutic target.

In summary, this transcriptional classification framework advances our understanding of PPGL heterogeneity and provides a theoretical foundation for designing targeted therapeutic strategies. By integrating molecular and immune features, it enables precision medicine approaches that have the potential to improve outcomes for patients with this challenging cancer type.

## Methods

### Patient cohorts

We conducted a retrospective analysis of patients diagnosed with PPGLs who underwent surgery at the First Hospital of Jilin University between October 2019 and March 2022. Formalin-fixed paraffin-embedded (FFPE) tumor samples from these patients were retrieved and reevaluated by two pathologists using hematoxylin and eosin staining and the original pathological reports. Patients with combined PPGLs or evidence of other tumors were excluded from the study. Clinical, pathological, and demographic data were obtained by reviewing electronic medical records. Follow-up information was gathered through regular visits or telephone interviews. RNA bulk sequencing was performed on FFPE samples from 87 patients with complete clinical data and confirmed PPGL diagnoses, along with 5 FFPE samples of normal adrenal tissue as controls. The study was conducted in accordance with the ethical standards of the Declaration of Helsinki and approved by the Ethics Committee of the First Hospital of Jilin University (23K101-001). Given its retrospective design, the need for informed consent was waived. Patient data were processed and analyzed anonymously to ensure confidentiality.

### Cell lines

Rat PPGL cell line PC-12 was obtained from the American Type Culture Collection (ATCC). The identity of the cell line has been verified. Authentication was performed using short tandem repeat profiling. All cell lines were routinely tested and confirmed to be free of mycoplasma contamination. None of the cell lines used are listed among the commonly misidentified cell lines maintained by the International Cell Line Authentication Committee. It was cultured in RPMI-1640 medium. Penicillin/streptomycin (#15140-122; Gibco) and 10% fetal bovine serum (#35-081-CV; Corning) were added to all media. All cell lines were cultured at 37°C with 5% $CO_2$.

### Mouse models

For CDX models, $1.0 \times 10^6$ PC-12 cell lines were subcutaneously injected into the flank of 6-week-old female BALB/c nude mice (Hafukang Co, Ltd, Beijing, China). Tumor size was measured every 2 days, and animal survival rate was recorded every day. Tumor size was calculated as $V = (L \times W^2)/2$, where $V$ is tumor volume, $L$ is the length of the tumor (longer diameter), and $W$ is the width. Mice with tumor size larger than 2.0 cm$^3$ at the longest axis were euthanized for ethical consideration. The Animal Care and Use Committee of First Hospital of Jilin University approved all procedures concerning mouse studies (11389).

## RNA extraction and gene expression profiling

Total RNA was successfully extracted from 87 FFPE samples using TRIzol and RNeasy MinElute Cleanup Kit (Invitrogen). RNA purity was assessed using the NanoDrop Spectrophotometer (Thermo Fisher Scientific, Waltham, USA). RNA integrity and concentration were measured with the RNA Nano 6000 Assay Kit of the Bioanalyzer 2100 system (Agilent Technologies, Palo Alto, CA, USA). Subsequently, mRNA libraries were created by using the NEB Next Ultra RNA Library Prep Kit (NEB, Beverly, MA, USA), following the manufacturer's protocol. Geneplus-2000 sequencing platform (Geneplus, Beijing, China) was utilized to sequence the constructed RNA-seq libraries. The sequencing reads containing adapter sequences and low-quality reads were removed to obtain high-quality reads. Reads passing quality control were aligned to the human genome hs37d5 using STAR software (*Dobin et al., 2013*), and transcripts were assembled using StringTie2 (*Kovaka et al., 2019*).

## IHC staining

IHC staining was performed to evaluate the expression of ANGPT2, PCSK1N, and GPX3 in FFPE tissue sections. Tissue samples were first baked at 65°C for 15 min. Then deparaffinize the sections in xylene (twice, 5 min each) and rehydrate through graded ethanol solutions (100%, 95%, and 70%, 5 min each). Antigen retrieval was conducted by heating the slides in Tris-EDTA buffer pH 9.0 (MVS-0098, MXB) at 95°C for 2 min using a pressure cooker. The slides were then allowed to cool to room temperature and washed with phosphate-buffered saline (PBS).

To block endogenous peroxidase activity, the sections were incubated with 3% hydrogen peroxide solution (BCCK5591, Sigma-Aldrich) for 20 min, and then washed with PBS. Non-specific antibody binding was minimized by pre-incubating the slides with 10% goat serum (C0265, Beyotime) for 1 hr at room temperature. The tissue sections were then incubated overnight at 4°C with primary antibodies targeting ANGPT2, PCSK1N, and GPX3 (proteintech-ANGPT2-24613-1-AP, abnova-PCSK1N-H00027344-M02, and Proteintech-GPX3-13947-1-AP).

After washing with PBS, the sections were incubated with HRP signal enhancement solution (PV-9001, ORIGENE) for 20 min and washed, then incubated with enhanced enzyme-labeled goat anti-rabbit IgG polymer (PV-9001, ORIGENE) for 20 min at room temperature. For chromogenic detection, the signal was developed using DAB (3,3'-diaminobenzidine) (DAB-4032, MXB) and counterstained with hematoxylin (G1004, Servicebio), differentiation solution (G1039, Servicebio), and bluing reagent (G1040, Servicebio), washed with distilled water. Slides were dehydrated through graded ethanol solutions, cleared in xylene, and mounted using neutral balsam (10004160, SCR).

Stained sections were examined under a light microscope (Eclipse E100, Nikon), and images were captured. Evaluate the staining pattern and intensity. Use Fiji ImageJ to measure the percentage of the stained area. Use the statistical software GraphPad Prism to perform a *T*-test.

## Antibodies and immunoblotting

Anti-ANGPT2 (Abcam, ab259823) antibodies were used for immunoblotting assay. Cells were lysed in lysis buffer supplemented with 1 mM phenylmethylsulfonyl fluoride. Cell lysates in SDS loading buffer were boiled for 15 min and then analyzed with SDS–PAGE (Bio-Rad). The samples were then transferred to polyvinylidene fluoride membranes (MilliporeSigma), which were then blocked with 2% milk in PBST buffer (phosphate-buffered saline with 0.05% Tween 20) for 1 hr before incubation with primary antibodies for 2 hr at room temperature or at 4°C overnight. After a wash with PBST, the membranes were incubated with a secondary antibody for 1 hr, and then washed three times with PBST. The samples were detected by adding SuperSignal West Femto Maximum Sensitivity substrate (NCM Biotech, P10300).

## Weighted gene co-expression network analysis

Overview and rationale. We applied WGCNA to identify modules of genes that show coordinated expression across samples and to relate these modules to study traits. Conceptually, WGCNA transforms pairwise gene–gene correlations into a weighted network, detects modules (clusters) of highly co-expressed genes, summarizes each module by its module eigengene (first principal component of the standardized expression matrix for that module), and correlates module eigengenes with external traits to prioritize biology-linked modules and hub genes.

Preprocessing. Raw counts were normalized and variance-stabilized ($\log_2$(TPM + 1) with library-size correction). Lowly expressed genes were filtered. Outlier samples were assessed by hierarchical clustering and removed if necessary.

Network construction and soft-threshold selection. We computed pairwise gene correlations and raised the absolute correlations to a soft-thresholding power $\beta$ to emphasize strong connections while preserving continuous weights. We selected $\beta$ = [value] using the pickSoftThreshold procedure, targeting a scale-free topology fit $R^2 \geq 0.8$ and mean connectivity within a reasonable range. A signed network (networkType="signed"), which preserves correlation direction, was used to focus modules on positively co-regulated genes.

Module detection and merging. Adjacency matrices were converted to a topological overlap matrix (TOM) to capture shared-neighbor structure. Genes were hierarchically clustered on TOM-based dissimilarity, and initial modules were identified using dynamic tree cutting. Highly similar modules were merged based on eigengene correlation. For large gene sets, we used blockwiseModules to ensure computational scalability.

The bulk-seq data of 87 patients were analyzed by WGCNA using the R package WGCNA, which was divided into three subtypes: C1, C2, and C3. Then, the clustering dendrogram and gene module heatmap of the genes were constructed to observe the significant gene modules of each phenotype. At the same time, the correlation analysis heatmap between the modules was constructed. The parameters were set as follows: minModuleSize = 15 and softPower = 7.

## Analysis of immune infiltration using xCell

The immune infiltration in tumor samples was assessed using the R package xCell to predict the relative proportions of 19 immune and stromal cell types within the TME. The final results were visualized using a heatmap generated by the R package pheatmap, displaying significant differences in cell enrichment across various samples. Violin plots were utilized to illustrate the differences in immune and stromal cell distributions among the three tumor subtypes in the PPGL cohort and TCGA datasets.

## Gene mutation analysis

For Gene Mutation Landscape, mutation profiles of each sample of TCGA data for genes related to PCPG were analyzed using the R package maftools. A waterfall plot of the gene mutation landscape was generated using the oncoplot function. For TMB, MSI, and mutant allele tumor heterogeneity (MATH) analysis, TCGA data was utilized, and the R package maftools was employed to read and calculate the MSI, MATH, and TMB scores. The MSI, MATH, and TMB scores across the three subtypes were compared to observe the mutation patterns in the three subtypes.

## Single-cell nuclear transcriptome sequencing (snRNA-seq)

For analysis of snRNA-seq data, single-cell nucleus RNA sequencing datasets were downloaded from the European Genome-Phenome Archive (EGA) under accession code EGAS00001005861. The snRNA-seq data was analyzed using the R package Seurat v4.4.1. The scRNA-seq data underwent filtration to incorporate genes expressed in no less than 3 cells, as well as cells that exhibited gene expression between 200 and 5000 genes, with mitochondrial transcripts comprising less than 5%. Given patient-driven clustering observed in the analysis of immune populations, we applied the SCTransform v2 method to correct the batch effects between different patients. To ascertain that SCTransform v2 solely addressed technical discrepancies and not biological variations, we manually compared the resulting clusters and their associated markers to those obtained from a single library cluster analysis. After normalizing the data, dimensionality reduction and clustering were performed, with unsupervised clustering analysis used to identify different cell subpopulations. The predominant cellular subpopulations within the immune compartments were delineated based on cluster-averaged gene expression levels of the subsequent gene markers: (1) chromaffin cells, PCPG and ATPG: *TH* and *CHGA*, (2) adrenocortical cells: *STAR*, (3) ECs: *FLT1*, (4) SCLCs: *SOX10*, (5) fibroblasts: *PDGFRB*, (6) myeloid cells: *MSR1*, (7) T/NK cells: *CD2*, (8) mast cells: *HDC*, and (9) B cells: *CD79A*. For in-depth investigation of immune cell populations, the dataset was narrowed down to exclusively include these specific cell types. Subsequently, the data underwent comprehensive processing from its raw form, following the aforementioned methodology. The

DimPlot function was used to visualize UMAP plots for all cell types, infiltrated immune cells, and different subtypes. The ggplot function was utilized to create stacked bar plots of cell type proportions.

### CellChat analysis

CellChat is an R package designed for inference, analysis, and visualization of cell–cell communication from single-cell and spatially resolved transcriptomics. CellChat aims to enable users to identify and interpret cell–cell communication within an easily interpretable framework, with the emphasis on clear, attractive, and interpretable visualizations.

### Pseudotime and enrichment analysis (monocle2)

The R package of monocle2 was used to perform the diffusion pseudotime analysis. Normalized data computed in Seurat was directly transferred to monocle2. The differential gene analysis function in monocle2 was used to identify genes with significant changes between C1, C2, C3, and normal subtypes ($q$-value <0.05), which were then utilized as input for temporal ordering of cells along the differentiation trajectory. The final results were subsequently transferred back to Seurat. Then, we performed KEGG and GO enrichment analysis by using the clusterProfiler package (v3.18.1).

### Survival analysis

The data for survival analysis was derived from PPGL cohort. The R packages survival and survminer were utilized to plot Kaplan–Meier survival curves. The survfit function was applied to fit the survival model. Ultimately, the ggsurvplot was employed to plot the survival curves, depicting the survival status of patients in different subtypes.

### Software and versions

Seurat (v4.4.0), sctransform (v0.4.2), CellChat (v2.2.0), monocle (v2.36.0; monocle2), pheatmap (v1.0.13), clusterProfiler (v4.16.0), survival (v3.8.3), and ggplot2 (v3.5.2).

## Acknowledgements

National Natural Science Foundation of China (grant numbers 82172690 and 82472799) to Lingyu Li.

## Additional information

### Funding

| Funder | Grant reference number | Author |
| --- | --- | --- |
| National Natural Science Foundation of China | 82172690 | Lingyu Li |
| National Natural Science Foundation of China | 82472799 | Lingyu Li |

The funders had no role in study design, data collection, and interpretation, or the decision to submit the work for publication.

### Author contributions

Yang Liu, Data curation, Software, Investigation, Visualization, Writing – original draft, Project administration; Xu Yan, Fengrui Nan, Data curation; Yibo Zhang, Resources, Data curation; Zhenfu Gao, Software; Siyu Shi, Formal analysis; Jingyun Chen, Investigation; Lingyu Li, Conceptualization, Resources, Project administration, Writing - review and editing

### Author ORCIDs
Yang Liu (iD) https://orcid.org/0000-0001-7968-3403
Lingyu Li (iD) https://orcid.org/0000-0001-9920-6508

## Ethics

The study was conducted in accordance with the ethical standards of the Declaration of Helsinki and approved by the Ethics Committee of the First Hospital of Jilin University (23K101-001). Given its retrospective design, the need for informed consent was waived. Patient data were processed and analyzed anonymously to ensure confidentiality.

For CDX models, $1.0 \times 10^6$ PC-12 cell lines were subcutaneously injected into the flank of 6-week-old female BALB/c nude mice (Hafukang Co, Ltd, Beijing, China). Tumor size was measured every 2 days, and animal survival rate was recorded every day. Tumor size was calculated as $V = (L \times W^2)/2$, where V is tumor volume, L is the length of the tumor (longer diameter), and W is the width. Mice with tumor size larger than $2.0 \text{ cm}^3$ at the longest axis were euthanized for ethical consideration. The Animal Care and Use Committee of First Hospital of Jilin University approved all procedures concerning mouse studies (11389).

Reviewer #1 (Public review): https://doi.org/10.7554/eLife.107108.3.sa1
Reviewer #2 (Public review): https://doi.org/10.7554/eLife.107108.3.sa2
Author response https://doi.org/10.7554/eLife.107108.3.sa3

# Additional files

## Supplementary files

MDAR checklist

## Data availability

The raw RNA-seq data generated in this study have been deposited in ArrayExpress under accession number E-MTAB-16109. We gratefully acknowledge Prof. Richard W Tothill for providing the single-nucleus RNA-seq dataset, which has been deposited in the European Genome-Phenome Archive (EGA) under accession EGAD00001008403. The immunohistochemistry (IHC) image dataset generated in this study has now been deposited and made publicly available in the BioStudies BioImage Archive under accession number S-BIAD2381.

The following datasets were generated:

| Author(s) | Year | Dataset title | Dataset URL | Database and Identifier |
|---|---|---|---|---|
| Liu Y, Yan X, Zhang Y, Gao Z, Nan F, Shi S, Chen J, Li L | 2025 | Transcriptional Subtypes on Immune Microenvironment and Predicting Postoperative Recurrence and Metastasis in Pheochromocytoma and Paraganglioma | https://www.ebi.ac.uk/biostudies/arrayexpress/studies/E-MTAB-16109 | ArrayExpress, E-MTAB-16109 |
| Liu Y | 2025 | Transcriptional Subtypes on Immune Microenvironment and Predicting Postoperative Recurrence and Metastasis in Pheochromocytoma and Paraganglioma | https://doi.org/10.6019/S-BIAD2381 | BioImage Archive, 10.6019/S-BIAD2381 |

The following previously published dataset was used:

| Author(s) | Year | Dataset title | Dataset URL | Database and Identifier |
|---|---|---|---|---|
| Zethoven M, Martelotto L, Pattison A, Bowen B, Balachander S, Flynn A, et al | 2022 | Single-nuclei and bulk-tissue geneexpression analysis of pheochromocytoma and paraganglioma links disease subtypes with tumor microenvironment | https://www.ega-archive.org/datasets/EGAD00001008403 | European Genome-Phenome Archive, EGAD00001008403 |

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
