## [Editor Report · eLife Assessment]

This is a **valuable** study describing transcriptome-based pheochromocytoma and paraganglioma (PPGL) subtypes and exploring the mutations, immune correlates and disease progression of cases in each subtype. The cohort is a reasonable size and a second cohort is included from the Cancer Genome Atlas (TCGA). One of the key premises of the study is that identification of driver mutations in PPGL is not complete and that compromises characterisation for prognostic purposes. This is a **solid** starting point on which to base characterisation using different methods.

---

## [Referee Report · Reviewer #1 (Public review)]

This study presents an exploration of PPGL tumour bulk transcriptomics and identifies three clusters of samples (labeled as subtypes C1-C3). Each subtype is then investigated for the presence of somatic mutations, metabolism-associated pathway and inflammation correlates, and disease progression.

The proposed subtype descriptions are presented as an exploratory study. The proposed potential biomarkers from this subtype are suitably caveated and will require further validation in PPGL cohorts together with mechanistic study.

The first section uses WGCNA (a method to identify clusters of samples based on gene expression correlations) to discover three transcriptome-based clusters of PPGL tumours using a new cohort of n=87 PPGL samples from various locations in the body.

The second section inspects a previously published snRNAseq dataset, assigning the published samples to subtypes C1-C3 using a pseudo-bulk approach.

The tumour samples are obtained from multiple locations in the body, summarised in Fig1A. It will be important to see further investigation of how the sample origin is distributed among the C1-C3 clusters, and whether there is a sample-origin association with mutational drivers and disease progression.

Comments on revisions:

In SupplFile3 (pdf) - please correct the table format. The contents are obscured due to the narrowness of the table columns.

Deposit the new RNAseq data (N=87 cases, N=5 controls) in an appropriate repository; see "Data on human genotypes and phenotypes" at https://elife-rp.msubmit.net/html/elife-rp_author_instructions.html#dataavailability

---

## [Referee Report · Reviewer #2 (Public review)]

Summary:

A study that furthers the molecular definition of PPGL (where prognosis is variable) and provides a wide range of sub-experiments to back up the findings. One of the key premises of the study is that identification of driver mutations in PPGL is incomplete and that compromises characterisation for prognostic purposes. This is a reasonable starting point on which to base some characterisation based on different methods.

Strengths:

The cohort is a reasonable size, and a useful validation cohort in the form of TCGA is used. Whilst it would be resource-intensive (though plausible given the rarity of the tumour type) to perform RNAseq on all PPGL samples in clinical practice, some potential proxies are proposed.

Weaknesses:

Performance of some of the proxy markers for transcriptional subtype is not presented.

Limited prognostic information available.

Comments on revisions:

Having reviewed the responses to my comments and associated revisions, I am satisfied that they have been addressed.

---

## [Author Response]

The following is the authors’ response to the original reviews.

**Reviewer #1 (Public Review):**
This study presents an exploration of PPGL tumour bulk transcriptomics and identifies three clusters of samples (labeled as subtypes C1-C3). Each subtype is then investigated for the presence of somatic mutations, metabolism-associated pathways and inflammation correlates, and disease progression. The proposed subtype descriptions are presented as an exploratory study. The proposed potential biomarkers from this subtype are suitably caveated and will require further validation in PPGL cohorts together with a mechanistic study.The first section uses WGCNA (a method to identify clusters of samples based on gene expression correlations) to discover three transcriptome-based clusters of PPGL tumours. The second section inspects a previously published snRNAseq dataset, and labels some of the published cells as subtypes C1, C2, C3 (Methods could be clarified here), among other cells labelled as immune cell types. Further details about how the previously reported single-nuclei were assigned to the newly described subtypes C1-C3 require clarification.

Thank you for your valuable suggestion. In response to the reviewer’s request for further clarification on “how previously published single-nuclei data were assigned to the newly defined C1-C3 subtypes,” we have provided additional methodological details in the revised manuscript (lines 103-109). Specifically, we aggregated the single-nucleus RNA-seq data to the sample level by summing gene counts across nuclei to generate pseudo-bulk expression profiles. These profiles were then normalized for library size, log-transformed (log1p), and z-scaled across samples. Using genesets scores derived from our earlier WGCNA analysis of PPGLs, we defined transcriptional subtypes within the Magnus cohort (Supplementary Figure. 1C). We further analyzed the single-nucleus data by classifying malignant (chromaffin) nuclei as C1, C2, or C3 based on their subtype scores, while non-malignant nuclei (including immune, stromal, endothelial, and others) were annotated using canonical cell-type markers (Figure. 4A).

The tumour samples are obtained from multiple locations in the body (Figure 1A). It will be important to see further investigation of how the sample origin is distributed among the C1C3 clusters, and whether there is a sample-origin association with mutational drivers and disease progression.

Thank you for your valuable suggestion. In the revised manuscript (lines 74-79), Figure. 1A, Table S1 and Supplementary Figure. 1A, we harmonized anatomic site annotations from our PPGL cohort and the TCGA cohort and analyzed the distribution of tumor origin (adrenal vs extra-adrenal) across subtypes. The site composition is essentially uniform across C1-C3— approximately 75% pheochromocytoma (PC) and 25% paraganglioma (PG)—with only minimal variation. Notably, the proportion of extra-adrenal origin (paraganglioma origin) is slightly higher in the C1 subtype (see Supplementary Figure 1A), which aligns with the biological characteristics of tumors from this anatomical site, which typically exhibit more aggressive behavior.

**Reviewer #2 (Public Review):**
A study that furthers the molecular definition of PPGL (where prognosis is variable) and provides a wide range of sub-experiments to back up the findings. One of the key premises of the study is that identification of driver mutations in PPGL is incomplete and that compromises characterisation for prognostic purposes. This is a reasonable starting point on which to base some characterisation based on different methods. The cohort is a reasonable size, and a useful validation cohort in the form of TCGA is used. Whilst it would be resource-intensive (though plausible given the rarity of the tumour type) to perform RNA-seq on all PPGL samples in clinical practice, some potential proxies are proposed.

We sincerely thank the reviewer for their positive assessment of our study’s rationale. We fully agree that RNA sequencing for all PPGL samples remains resource-intensive in current clinical practice, and its widespread application still faces feasibility challenges. It is precisely for this reason that, after defining transcriptional subtypes, we further focused on identifying and validating practical molecular markers and exploring their detectability at the protein level.

In this study, we validated key markers such as ANGPT2, PCSK1N, and GPX3 using immunohistochemistry (IHC), demonstrating their ability to effectively distinguish among molecular subtypes (see Figure. 5). This provides a potential tool for the clinical translation of transcriptional subtyping, similar to the transcription factor-based subtyping in small cell lung cancer where IHC enables low-cost and rapid molecular classification.

It should be noted that the subtyping performance of these markers has so far been preliminarily validated only in our internal cohort of 87 PPGL samples. We agree with the reviewer that largerscale, multi-center prospective studies are needed in the future to further establish the reliability and prognostic value of these markers in clinical practice.

The performance of some of the proxy markers for transcriptional subtype is not presented.

We agree with your comment regarding the need to further evaluate the performance of proxy markers for transcriptional subtyping. In our study, we have in fact taken this point into full consideration. To translate the transcriptional subtypes into a clinically applicable classification tool, we employed a linear regression model to compare the effect values (β values) of candidate marker genes across subtypes (Supplementary Figure. 1D-F). Genes with the most significant β values and statistical differences were selected as representative markers for each subtype.

Ultimately, we identified ANGPT2, PCSK1N, and GPX3—each significantly overexpressed in subtypes C1, C2, and C3, respectively, and exhibiting the most pronounced β values—as robust marker genes for these subtypes (Figure. 5A and Supplementary Figure. 1D-F). These results support the utility of these markers in subtype classification and have been thoroughly validated in our analysis.

There is limited prognostic information available.

Thank you for your valuable suggestion. In this exploratory revision, we present the available prognostic signal in Figure. 5C. Given the current event numbers and follow-up time, we intentionally limited inference. We are continuing longitudinal follow-up of the PPGL cohort and will periodically update and report mature time-to-event analyses in subsequent work.

**Reviewer #1 (Recommendations for the authors):**
There is no deposition reference for the RNAseq transcriptomics data. Have the data been deposited in a suitable data repository?

Thank you for your valuable suggestion. We have updated the Data availability section (lines 508–511) to clarify that the bulk-tissue RNA-seq datasets generated in this study are available from the corresponding author upon reasonable request.

In the snRNAseq analysis of existing published data, clarify how cells were labelled as "C1", "C2", "C3", alongside cells labelled by cell type (the latter is described briefly in the Methods).

Thank you for your valuable suggestion. In response to the reviewer’s request for further clarification on “how previously published single-nuclei data were assigned to the newly defined C1-C3 subtypes,” we have provided additional methodological details in the revised manuscript (lines 103-109). Specifically, we aggregated the single-nucleus RNA-seq data to the sample level by summing gene counts across nuclei to generate pseudo-bulk expression profiles. These profiles were then normalized for library size, log-transformed (log1p), and z-scaled across samples. Using genesets scores derived from our earlier WGCNA analysis of PPGLs, we defined transcriptional subtypes within the Magnus cohort (Supplementary Figure. 1C). We further analyzed the single-nucleus data by classifying malignant (chromaffin) nuclei as C1, C2, or C3 based on their subtype scores, while non-malignant nuclei (including immune, stromal, endothelial, and others) were annotated using canonical cell-type markers (Figure. 4A).

Package versions should be included (e.g., CellChat, monocle2).

We greatly appreciate your comments and have now added a dedicated “Software and versions” subsection in Methods. Specifically, we report Seurat (v4.4.0), sctransform (v0.4.2), CellChat (v2.2.0), monocle (v2.36.0; monocle2), pheatmap (v1.0.13), clusterProfiler (v4.16.0), survival (v3.8.3), and ggplot2 (v3.5.2) (lines 514-516). We also corrected a typographical error (“mafools” → “maftools”) (lines 463).

**Reviewer #2 (Recommendations for the authors):**
It would be helpful to provide a little more detail on the clinical composition of the cohort (e.g., phaeo vs paraganglioma, age, etc.) in the text, acknowledging that this is done in Figure 1.

Thank you for your valuable suggestion. In the revision, we added Table S1 that provides a detailed summary of the clinical composition of the PPGL cohort. Specifically, we report the numbers and proportions (Supplementary Figure. 1A) of pheochromocytoma (PC) versus paraganglioma (PG), further subclassifying PG into head and neck (HN-PG), retroperitoneal (RPPG), and bladder (BC-PG).

How many of each transcriptional subtype had driver mutations (germline or somatic)? This is included in the figures but would be worth mentioning in the text. Presumably, some of these may be present but not detected (e.g., non-coding variants), and this should be commented on. It is feasible that if methods to detect all the relevant genomic markers were improved, then the rate of tumours without driver mutations would be less and their prognostic utility would be more comprehensive.

Thank you for your valuable suggestion. In the revision (lines 113–116), we now report the prevalence of driver mutations (germline or somatic) overall and by transcriptional subtype. We analyzed variant data across 84 PPGL-relevant genes from 179 tumors in the TCGA cohort and 30 tumors in Magnus’s cohort (Fig. 2A; Table S2). High-frequency genes were consistent with known biology—C1 enriched for [e.g., VHL/SDHB], C2 for [e.g., RET/HRAS], and C3 for [e.g., SDHA/SDHD]. We also note that a subset of tumors lacked an identifiable driver, which likely reflects current assay limitations (e.g., non-coding or structural variants, subclonality, and purity effects). Broader genomic profiling (deep WGS/long-read, RNA fusion, methylation) would be expected to reduce the “driver-negative” fraction and further enhance the prognostic utility of these classifiers.

ANGPT2 provides a reasonable predictive capacity for the C1 subtype as defined by the ROC AUC. What was the performance of the PCSK1N and GPX3 as markers of the other subtypes?

We agree with your comment regarding the need to further evaluate the performance of proxy markers for transcriptional subtyping, and we have supplemented the analysis with ROC and AUC values for two additional parameters (Author response image 1 , see below). Furthermore, in our study, we have in fact taken this point into full consideration. To translate the transcriptional subtypes into a clinically applicable classification tool, we employed a linear regression model to compare the effect values (β values) of candidate marker genes across subtypes (Supplementary Figure. 1D-F). Genes with the most significant β values and statistical differences were selected as representative markers for each subtype.

Ultimately, we identified ANGPT2, PCSK1N, and GPX3—each significantly overexpressed in subtypes C1, C2, and C3, respectively, and exhibiting the most pronounced β values—as robust marker genes for these subtypes (Figure. 5A and Supplementary Figure. 1D-F). These results support the utility of these markers in subtype classification and have been thoroughly validated in our analysis.

**Author response image 1. sa3fig1:** Extended Data Figure A-B. (A) The ROC curve illustrates the diagnostic ability to distinguish PCSK1N expression in PPGLs, specifically differentiating subtype C2 from non-C2 subtypes. The red dot indicates the point with the highest sensitivity (93.1%) and specificity (82.8%). AUC, the area under the curve. (B) The ROC curve illustrates the diagnostic ability to distinguish GPX3 expression in PPGLs, specifically differentiating subtype C3 from non-C3 subtypes. The red dot indicates the point with the highest sensitivity (83.0%) and specificity (58.8%). AUC, the area under the curve.

In the discussion, I think it would be valuable to summarise existing clinical/molecular predictors in PPGL and, acknowledging that their performance may be limited, compare them to the potential of these novel classifiers.

Thank you for your valuable suggestion. We have added a concise overview of established clinical and molecular predictors in PPGL and compared them with the potential of our transcriptional classifiers. The new paragraph (Discussion, lines 315–338) now reads:

“Compared to existing clinical and molecular predictors, risk assessment in PPGL has long relied on the following indicators: clinicopathological features (e.g., tumor size, non-adrenal origin, specific secretory phenotype, Ki-67 index), histopathological scoring systems (such as PASS/GAPP), and certain genetic alterations (including high-risk markers like SDHB inactivation mutations, as well as susceptibility gene mutations in ATRX, TERT promoter, MAML3, VHL, NF1, among others). Although these metrics are highly actionable in clinical practice, they exhibit several limitations: first, current molecular markers only cover a subset of patients, and technical constraints hinder the detection of many potentially significant variants (e.g., non-coding mutations), thereby compromising the comprehensiveness of prognostic evaluation; second, histopathological scoring is susceptible to interobserver variability; furthermore, the lack of standardized detection and evaluation protocols across institutions limits the comparability and generalizability of results. Our transcriptomic classification system—comprising C1 (pseudohypoxic/angiogenic signature), C2 (kinase-signaling signature), and C3 (SDHx-related signature)—provides a complementary approach to PPGL risk assessment. These subtypes reflect distinct biological backgrounds tied to specific genetic alterations and can be approximated by measuring the expression of individual genes (e.g., ANGPT2, PCSK1N, or GPX3). This study demonstrates that the classifier offers three major advantages: first, it accurately distinguishes subtypes with coherent biological features; second, it retains significant predictive value even after adjusting for clinical covariates; third, it can be implemented using readily available assays such as immunohistochemistry. These findings suggest that integrating transcriptomic subtyping with conventional clinical markers may offer a more comprehensive and generalizable risk stratification framework. However, this strategy would require validation through multi-center prospective studies and standardization of detection protocols.”

A little more explanation of the principles behind WGCNA would be useful in the methods.

We are grateful for your comments. We have expanded the Methods to briefly explain the principles of WGCNA (lines 426-454). In short, WGCNA constructs a weighted coexpression network from normalized gene expression, identifies modules of tightly co-expressed genes, summarizes each module by its eigengene (the first principal component), and then correlates module eigengenes with phenotypes (e.g., transcriptional subtypes) to highlight biologically meaningful gene sets and candidate hub genes. We now specify our preprocessing, choice of softthresholding power to approximate scale-free topology, module detection/merging criteria, and the statistics used for module–trait association and downstream gene-set scoring.

On line 234, I think the figure should be 5C?

We greatly appreciate your comments and Correct to Figure 5C.